# Multispectral Facial Recognition in the Wild

**DOI:** 10.3390/s22114219

**Published:** 2022-06-01

**Authors:** Pedro Martins, José Silvestre Silva, Alexandre Bernardino

**Affiliations:** 1Military Electrical and Computer Engineering, Portuguese Military Academy, Rua Gomes Freire, 1169-203 Lisbon, Portugal; pedro.roque.martins@tecnico.ulisboa.pt; 2Instituto Superior Técnico, Universidade de Lisboa, 1049-001 Lisbon, Portugal; alex@isr.tecnico.ulisboa.pt; 3Military Academy Research Center (CINAMIL), Rua Gomes Freire, 1169-203 Lisbon, Portugal; 4Laboratory for Instrumentation, Biomedical Engineering and Radiation Physics (LIBPhys-UC), 3000-370 Coimbra, Portugal; 5Institute for Systems and Robotics (ISR), 1049-001 Lisbon, Portugal

**Keywords:** deep neural networks, multispectral imaging, face recognition, in the wild

## Abstract

This work proposes a multi-spectral face recognition system in an uncontrolled environment, aiming to identify or authenticate identities (people) through their facial images. Face recognition systems in uncontrolled environments have shown impressive performance improvements over recent decades. However, most are limited to the use of a single spectral band in the visible spectrum. The use of multi-spectral images makes it possible to collect information that is not obtainable in the visible spectrum when certain occlusions exist (e.g., fog or plastic materials) and in low- or no-light environments. The proposed work uses the scores obtained by face recognition systems in different spectral bands to make a joint final decision in identification. The evaluation of different methods for each of the components of a face recognition system allowed the most suitable ones for a multi-spectral face recognition system in an uncontrolled environment to be selected. The experimental results, expressed in Rank-1 scores, were 99.5% and 99.6% in the TUFTS multi-spectral database with pose variation and expression variation, respectively, and 100.0% in the CASIA NIR-VIS 2.0 database, indicating that the use of multi-spectral images in an uncontrolled environment is advantageous when compared with the use of single spectral band images.

## 1. Introduction

The sense of sight allows us to observe dangers, identify objects, and recognize people. This last task is fundamental for human beings as social beings. It enables us to differentiate the level of trust someone can give to a specific person, with this being at the base of the construction of communities. Such is the importance of this task that it has become one of the main topics of research with the emergence of machine learning, thus allowing machines to incorporate this biological capacity.

Multi-spectral images have several military applications, from detection of camouflaged people [1], classification of vegetation types in military regions [2], landmine detection [3] and face recognition [4]. The current face recognition systems operating in the visible (VIS) domain have reached a significant level of maturity. It is possible to observe their wide use nowadays, from security mechanisms to unlocking electronic devices such as smartphones and personal computers to population control systems [5].

However, most current face recognition systems [6] require the cooperation of the user to ensure that pictures are taken in favorable conditions (frontal postures, good illumination, no occlusion) and have trouble dealing with uncontrolled scenarios. Uncontrolled environment scenarios, such as riots and violent demonstrations, can often be used by criminals and terrorist cell members to move around and cause damage to Homeland Security, as this type of environment adds difficulty to their detection. The uncontrolled environment is mainly characterized by a variety of lighting, pose, facial expressions and the existence of occlusions [5]. These features are challenges to face recognition systems due to the multiple intrapersonal variations they provide, making it difficult to correctly identify an individual’s identity based on a collaborative image of the individual.

This work has as its main objective the development of a multi-spectral face recognition system in an uncontrolled environment. To achieve this goal, the solutions used by current recognition systems and the evaluation of the benefits of using multi-spectral images are explored. The developed face recognition system is evaluated in public multi-spectral image datasets with pose and expression variability.

This paper is organized into six sections. The Introduction section describes the motivation for the work, the objectives and the structure of the paper. The Background section explains important concepts, such as how a face recognition system works, what multispectral images are and what their advantages are. The Related Work section presents the study of the art of multispectral face recognition methods in an uncontrolled environment and of public multispectral databases. The Methodology section defines the proposed method in order to achieve the objectives. The Results and Discussion section describes the multispectral databases used, several experiments are also performed with the various modules proposed in the methodology, each experiment is accompanied by its respective analysis and discussion. The Conclusions section presents the conclusions of this work, thus consolidating the proposed objectives.

## 2. Background

### 2.1. Face Recognition

In general, a face recognition system is described in several phases. The first phase consists of acquiring the facial images and pre-processing them, such as locating the faces and cropping them. In a second phase, the features are extracted from the facial image, for instance, the position of facial landmarks, eye distance or even the face tones. Finally, these features are used in a classifier for identification or verification purposes.

Face recognition can be performed in a controlled or uncontrolled environment. The controlled environment, also known as consent recognition, is one in which the user cooperates in the recognition by facilitating it through correct and static posture in a place with good lighting. In the uncontrolled environment, recognition is dynamic, without the user cooperating in acquiring an image, making the face recognition process very difficult due to the diversity of the surrounding environment (e.g., low visibility), facial poses and expressions.

### 2.2. Multispectral Imaging in an Uncontrolled Environment

The databases of the VIS domain and the use of image synthesizers, which generate multiple poses and facial expressions from the obtained images, have allowed the difficulties associated with the variety of poses and facial expressions to be circumvented. However, two points have proved more difficult to overcome: the change of illumination and occlusions. This has led to the use of multiple spectral bands, with particular emphasis on the infrared (IR) spectral band, which can acquire images in environments with little or no brightness and overcome occlusions such as smoke and fog. In short, multispectral analysis allows a face recognition system to extract facial features that would be impossible to obtain with images from the VIS spectral band.

The IR bands can be categorized according to several spectral bands [7]. The active bands are the near-infrared (NIR) and short-wavelength infrared (SWIR). To acquire images in these bands, the object must receive illumination, even if scarce, because it is through reflection that the image is acquired. Such a fact means these images are commonly used in night vision devices. The NIR band allows the difficulties posed by the variation of illumination to be overcome, while the SWIR has the advantage of obtaining images through smoke and fog. The passive bands are the mid-wavelength infrared (MWIR) and long-wavelength infrared (LWIR). Unlike the active bands, the passive bands allow us to acquire images using only the thermal radiation emitted by a body, commonly known as thermal images.

The use of IR images for automatic face recognition is not without challenges, as these images are sensitive to the emotional, physical and health conditions of the individual, as well as the surroundings, and do not serve as an absolute alternative to the use of the VIS spectrum, but rather as a complement [8]. Another difficulty arises from the low number of public databases with images from both spectral ranges and in an uncontrolled environment [9], which limit the creation of rich classification models and the ability to characterize the performance of those systems in realistic conditions.

## 3. Related Work

Multi-spectral face recognition in an uncontrolled environment can be subdivided into two areas. The first is face recognition in an uncontrolled environment, which is already challenging. The second is multi-spectral face recognition, i.e., using different spectral bands in face recognition. This section briefly reviews the progress made in these two areas.

### 3.1. Face Recognition in an Uncontrolled Environment

The uncontrolled environment, strongly characterized by pose-light-expression factors, emerges as a problem for current recognition systems. A significant step was taken towards solving this type of problem by introducing very large databases to train Deep Convolutional Neural Networks (DCNN) in combination with the emergence of image synthesis methods [5]. The two main image synthesis methods are: (i) one-to-many augmentation, which consists of generating different poses of a face from a canonical face image; (ii) many-to-one normalization, which consists of normalizing any pose of the face to a canonical face pose [5]. The use of Generative Adversarial Networks (GAN), introduced by Goodfellow et al. [10], is characterized by the use of a generator and a discriminator (see Figure 1). The generator is responsible for producing samples given an input image so that the discriminator cannot discern which of the samples is real and which is false.

Since their appearance in face normalization, with DR-GAN [11], GANs have taken the lead in solving the problem of pose and facial expression variation. As for one-to-many augmentation using GANs, as is the case with the DA-GAN network [12], their image production power also gives them an advantage compared to other algorithms.

Normalization of many-to-one images is an extreme image synthesis problem due to the pose differences of a face. Cao et al. [13] proposed HF-PIM, normalizing the face to a frontal pose through a texture fusion deformation procedure leveraging a dense matching field to interconnect the 2D and 3D surface spaces. Qian et al. [14] presented Face Normalization Module (FNM), which encodes images using a pre-trained network for feature extraction and generates realistic images.

One-to-many augmentation is another approach to achieve face recognition regardless of the pose. Tran et al. [15] synthesized different poses through 3D modeling and then trained a DCNN to perform face recognition with varied poses. The DA-GAN proposed by Zhao et al. [12] created 2D images through 3D modeling and then refined the obtained 2D images to be as realistic as possible, using a GAN to try to preserve the identity of the face. Thus, the DA-GAN network was also used to augment the training data.

### 3.2. Multispectral Face Recognition

The main multi-spectral face recognition methods can be characterized by three important features: Image Synthesis Methods, Fusion Methods and Loss Functions.

Fusion methods are subdivided into feature fusion and score fusion. In the first, a fusion of features from the different spectral bands of the facial image is performed, allowing the most relevant features to be extracted from the different bands and joining them in a vector. The second method combines the scores obtained from each classifier uni-band (e.g., a classifier operating only in the LWIR band and another operating only in the NIR band) [16].

The image synthesis methods allow an image of a spectral band to be transformed into another, helping to compare two images. The main advantage of image synthesis is that it enables an image to be passed from any spectral band to the VIS band, making it possible to use classifiers implemented to process images of the VIS spectrum [17]. One of the most recent works in this area synthesizes VIS images from NIR images using GANs [18].

Finally, all neural networks have cost functions for the training moment to update the network weights. However, certain cost functions have been proposed to proceed specifically to the classification of multi-spectral images. Examples of these cost functions are the Scatter Loss [19] and the Wasserstein Distance [20].

### 3.3. Gaps

Although several scientific works address multi-spectral face recognition, few of these demonstrate its power in an uncontrolled environment due to the limitations in current databases of multi-spectral face images. In existing datasets, the variations of conditions are not extreme, as they are usually semi-controlled environments and not *in the wild* (uncontrolled environment). For example, the most studied database in multi-spectral face recognition, CASIA NIR-VIS 2.0 [21], uses images in which the pose has few deviations from the frontal position, which does not reliably characterize the uncontrolled environment. Thus, the fact that these databases are incomplete (compared to those of the VIS band) is still a barrier to improving the capability of multi-spectral face recognition systems in an uncontrolled environment.

The present work proposes a system that integrates the capabilities of current face recognition systems in an uncontrolled environment in the VIS spectrum at the pose variation level and the capabilities of multi-spectral face recognition systems to surpass illumination variation.

## 4. Methodology

The proposed multi-spectral face recognition system consists of three tasks: Face Detection and Alignment, Image Synthesis and Face Recognition. In Figure 2, the general operation of the proposed face recognition system is shown, including the steps performed in each task.

In the initial phase of the system, it is necessary to acquire multi-spectral images, which can be obtained through mono-spectral equipment (collects the image in only one spectral band) or multi-spectral (collects the image in different spectral bands). After image acquisition, the Face Detection and Alignment module aims to obtain an aligned and centered facial image with predefined dimensions. To achieve this goal, it is necessary to detect the presence of human faces and then perform a face marking, detecting essential landmarks of the face, such as eyes and nose, allowing a correct alignment of the face and clipping around it. The following task is Image Synthesis, which aims to obtain a frontal facial image. The next task is Face Recognition, where facial image features are extracted through a CNN and a one-shot learning methodology is followed for the classification task, obtaining similarity scores for each spectral band. These scores are combined using a score fusion method, and the predicted identity is the one with the highest combined score.

### 4.1. Face Detection and Alignment

Face detection, in conjunction with face alignment, aims to detect the faces presented in the input image and identify facial landmarks so that faces are centered, aligned and equally sized. Since face detection algorithms detect faces in rectangular areas without rotating the image, a face landmark detection algorithm is needed to apply a rotation so that the face is aligned on the horizontal plan, using the imaginary eye line. Thus, the procedure of face detection and alignment module (see Figure 3) does the following: is given an image, identifies the different faces present, extracts the facial landmarks and processes the image to produce facial images where the face is centered and aligned.

The face detection algorithms explored in this work are based on SSD (single-shot multibox detector), a deep learning architecture for object detection [22]. The basic idea of the SSD is to generate scores for the presence of each object category in each predefined box and produce adjustments to the box to match the shape of the object. In this work, three SSD based methods are tested: (i) the S3FD algorithm [23], (ii) the facial detection deep neural network of OpenCV [24] and (iii) the DSFD algorithm [25]. The S3FD has contributions to better cope with scaling variations with a single deep network. The DSFD uses a feature enhancement module to extend the single-shot detector to a dual-shot detector, obtaining more robust and discriminable features.

As for the facial landmark detection algorithms, the DLIB library’s 68 landmark network, adapted from Khazemi and Sullivan [26], and Bulat’s 2D-FAN [27], also with 68 landmarks, were tested. The latter one uses an Hour-Glass [28] based architecture to estimate the human pose. Both networks receive an image of a person and produce, as output, the position of the different facial landmarks around the face.

All the algorithms addressed in this subsection were trained in databases that only contain images in the spectral band of the VIS. To achieve data normalization, it is necessary to (i) rotate the image to align the eye line with the horizontal, (ii) crop the image to center the face image, and (iii) resize the image so that all output images have the specified dimensions.

### 4.2. Image Synthesis

To overcome the problems related with image acquisition in an uncontrolled environment, such as variation in lighting, occlusions and changes of poses, a face normalization module is used. This module aims to synthesize (create) an image of a face with frontal pose and neutral expression from a non-frontal face image.

To exemplify the expected behavior, Figure 4 shows an input face image in a non-frontal pose, with which the image synthesis module produces a frontal face image. Thus, it is intended that the image acquired helps obtain the identity features present in the facial image. The models FNM [14] and FFWM [29] are analyzed.

FNM is a GAN with two new features. First, it uses a network specialized in obtaining facial features to build the generator and provide the ability to preserve facial identity. Second, facial discriminators are used to refine local textures. Their authors claim that this model produces a face in the canonical pose without expression, which directly improves the performance of a face recognition system.

The normalization method of the FFWM model consists of using a deformation module, aiming to synthesize realistic frontal images with illumination preservation. For frontal image synthesis, it presents a module responsible for reducing pose discrepancy at the facial features level, thus preserving more details of profile images. The FFWM model uses pairs of face images for the training phase: one with a non-frontal pose and another with a frontal pose of the same person in the same conditions. Differently, the FNM model uses non-pair face images, where the images are not of the same person.

### 4.3. Face Recognition

This last module aims to identify the person present in an input face image, following the flowchart presented in Figure 5. For this purpose, it is necessary to perform two tasks: feature extraction and classification.

The extraction of representative features from a facial image is performed through a version of Light CNN [30] with 29 convolutional layers (Light CNN-29). To use this network for feature extraction in spectra other than VIS, transfer learning is used. According to [31], several models for biometric recognition are based on transfer learning when the databases are limited. Thus, one should use the Light CNN-29 model with the weights obtained by training on the VIS databases and fine tune with the facial image databases in spectra other than the VIS. At the end of the feature extraction phase, *B* vectors of 256 dimensions are generated, with *B* being the number of spectral bands in which the facial image was acquired.

The classification process applied by the one-shot learning technique determines the degree of similarity of the feature set extracted from the input image with the features sets extracted from the images of each class present in the support set, which is constituted by one example per class. The similarity functions to be used are the Euclidean distance and the cosine similarity. After obtaining the similarity values for each identity in the different spectral bands, a fusion of the obtained scores is performed, inspired by [27]:(1)$$S_{ic}=\overset{B}{\underset{b=1}{\sum }}S_{ib}W_{b}$$
where *S_ic_* is the combined score for each identity *i* and *S_ib_* is the score obtained for each band *b* for each identity *i*. *W_b_* is the weight of each spectral band. The weights associated with each band are fixed, determined by the accuracy obtained when classifying with only that band. In this way, the band that usually obtains the most reliable similarity scores to classify will have a greater weight in the fusion of scores. The prediction is then made by choosing the identity *i* of the support set that has the highest combined similarity score:(2)$$prediction=max\left(S_{ic}\right)\;\forall i\in \left[1,\ldots ,N\right]$$

## 5. Results and Discussion

### 5.1. Databases

We performed both qualitative and quantitative evaluations of the proposed methods. These images are in the VIS, NIR and LWIR bands. Two multi-spectral databases were used for quantitative evaluation: TUFTS [9] and CASIA NIR-VIS 2.0 [19]. The TUFTS database has facial images in the VIS, NIR and LWIR bands of 113 people with different poses and different illumination conditions. The TUFTS database has different subsets, divided into TUFTS-Pose (facial images with nine different poses per individual, in visible, NIR and LWIR) and TUFTS-Exp (four facial images with different expressions and one with sunglasses per individual, in visible and LWIR) to study pose variation and expression variation separately. CASIA NIR VIS 2.0 comprises 17,489 facial images of 715 people in VIS and NIR spectral bands under different light conditions.

### 5.2. Metrics

The metrics used are Rank-1, Rank-5 and TAR@FAR = 0.001. When using a generic expression Rank-n, given an image of a face as input, the classifier obtains the n most probable identities, one of which is the correct identity. TAR (true accept rate) is defined as the percentage of faces that, compared to the corresponding gallery identity, are identified as matches, while FAR (false accept rate) is the percentage of incorrect identities to which a face is matched.

### 5.3. Face Detection and Alignment

#### 5.3.1. Face Detection

Regarding the qualitative results presented in Figure 6, all algorithms produced similar results in the VIS band. This was expected since they were all trained in databases of the spectral band of the VIS. In the LWIR spectral band, a failure of the OpenCV network was observed in the second facial pose, where it cannot detect any face. In addition, when OpenCV and S3FD detect the faces, there is a variation in the rectangle area compared to the VIS spectral band. The DSFD maintained the same results, which is a good indicator of its ability to extract characteristics even in the LWIR spectral band.

The quantitative results are presented in Table 1. It can be observed that the OpenCV network results are lower than the others, especially in infrared bands. Comparing results between the S3FD network and the DSFD, it is observed very similar results in the spectral band of the VIS and NIR. However, the results in LWIR are about 8 percentage points better. We observe that the DSFD maintains a very high accuracy for the different spectral bands, thus being the best network for face detection in a multispectral facial analysis system.

#### 5.3.2. Landmark Detection and Facial Alignment

The results for face landmark detection are shown in Figure 7 and Figure 8. For the more challenging poses, we can see that the DLIB network fails, even in the VIS band (right eye, in Figure 7c), as it tends to maintain the shape of a near-frontal face. One possible cause of this behavior is that the face landmark detection model was trained in a dataset without significant variations at the pose level. The DLIB network reveals even more difficulties in the spectral band of LWIR.

2D-FAN reveals a good extraction of landmarks in any of the poses, including the LWIR band, where the results are somewhat like those obtained in the VIS band (Figure 8). In the case of Figure 8n, although it looks like there was a total failure, it is possible to observe that the eyes are correctly identified. 2D-FAN, unlike DLIB, was trained on a database with pronounced pose variations (including profile images), which is the justification for achieving better results.

Given the previous considerations, we decided to use the 2D-FAN over the DLIB’s network due to two factors: (i) it shows better results with face pose variation, and (ii) it is the only one capable of producing positive results in the LWIR spectral band. After the face detection with DSFD and landmark face detection with 2D-FAN, the align, crop and resize phase took place, which aligned the imaginary eye line of all detected faces with the horizontal, centered the faces in the images, cropped them and resized to the same size, resulting in the results presented in Figure 9. The alignment effect is strongly noticeable on the rightmost facial image. This normalization of the facial images can help a multispectral face recognition system in an uncontrolled, where faces can be presented in several poses.

### 5.4. Image Synthesis

For all images used in the qualitative and quantitative evaluations, the images were previously processed to be properly centered, aligned and scaled. The FFWM model needs to receive the facial images with certain facial landmarks always in the same coordinates. Therefore, the face detection and alignment module provided by the authors of FFWM was used to obtain the results. The images used by the FNM model were processed by the face detection and alignment module developed by the authors of this work. The rightmost images used in the previous tasks were replaced by ones with a strong expression, to evaluate the capacity of the models to normalize expressions.

#### 5.4.1. Selecting the Best Model

In Figure 10, the results obtained by the FFWM are shown. One of the images of the dataset could not be detected by the module provided by the authors of FFWM (see Figure 10n). It is possible to see that the performance of FFWM has a sharp drop as it moves away from the VIS band. Analyzing only the spectral band of the VIS and the images with pose variation (Figure 10b,c), a suitable normalization of the pose in Figure 10c is present.

In Figure 10b, the FFWM produces a deformed face when the person looks upwards. The exclusive use of the Multi-PIE database [32] in training the FFWM means that it can only normalize the face where the pose varies along the horizontal plane.

The FNM presents more satisfactory results (see Figure 11) in the NIR spectral band, where the facial images are more realistic than those of the FFWM. It should be noted that with the FNM model, identities change, i.e., the person in the output face image appears to be different from the person in the input face image. However, the use of a face feature extractor by the FNM model allows the most relevant features in the output face image to be kept. It is also relevant to point out that the FNM normalizes pose and expression, eliminates face masks, as is the case of the surgical mask, and normalizes to the VIS spectral band. However, this normalization does not produce realistic results with the LWIR images due to the difference between the LWIR and VIS spectral bands.

Given the previous considerations, we decided to use the FNM instead of the FFWM due to two factors: (i) the FFWM requires a specific face detection and alignment module and that the face is perpendicular to the horizontal, while the FNM is more robust to pose variations in the input image; (ii) all images normalized by the FNM tend to maintain the face proportions, without deforming them, in the NIR and VIS spectral bands.

#### 5.4.2. Evaluation of Selected Model

Identification with and without the use of FNM was performed to verify its advantage. For this purpose, the Light CNN-29 was used for feature extraction, and the identification was performed based on the score obtained by cosine similarity.

The results presented in Table 2 show that, without using the FNM, the use of the NIR spectral band produces better results than the VIS band in all metrics analyzed. A possible explanation is that the images obtained in the NIR band are not so affected by the illumination variation (due to pose variation), thus not causing as many occlusions as in the VIS band. The results improve with the use of the FNM in the VIS and NIR spectral bands, with increases in performance in Rank-1 of 15.9% and 0.7%, respectively. In the remaining metrics, it is also observed better values with the use of the normalization model. This shows that the apparent identity change in the qualitative tests (see Figure 11) does not have a negative impact. The results in the LWIR spectral band indicate that using the FNM does not improve the performance in any of the metrics.

Due to FNM’s ability to normalize facial expression, tests were performed with TUFTS-Exp to verify whether normalization of expression allowed Light CNN-29 to extract more representative facial features. The results presented in Table 3 show that the sets of features extracted by Light CNN-29 without facial expression normalization are already representative enough, obtaining a Rank-1 of 99.6% in the VIS and 67.5% in the LWIR and a TAR@FAR = 0.001 of 99.4% in the VIS band and 57.0% in the LWIR band. The use of FNM impairs the feature extraction and consequently the results, especially in the LWIR spectral band, where FNM has more difficulties in generating realistic images. Analyzing the results obtained, the FNM model is used only to normalize facial images from the TUFTS-Pose database in the VIS and NIR spectral bands.

Table 4 presents the results obtained for Rank-1 with the variation of the quantized pose. The values achieved in the VIS band show a significant improvement in the Rank-1 metric with the use of the FNM, resulting in an increase from 77.5% to 97.7% with pose variations of 45° and from 43.3% to 87.4% with pose variations of 60. In the NIR, there is only an improvement when the pose variation is 60°, where the results go from 93.4% to 96.5%. The results obtained prove the ability of the FNM network regarding the pose normalization, where a higher pose variation results in a higher benefit of using it.

### 5.5. Face Recognition

#### 5.5.1. Network Training

For the training phase, and considering the results presented above, it was decided to make only one fine adjustment to the LWIR band feature extraction network, because the results obtained in this band are considerably lower, due to the network having been trained in the visible. Thus, the fine-tuning aims for the network to learn to extract more representative features from facial images in the LWIR spectral band. In order to train the Light CNN-29 with identities (people) different from the test ones, a last connected layer was added for training purposes and the LWIR spectral band images from the IRIS database [33] were used. This last layer is used as the input of the softmax cost function and is simply set to the number of training set identities, as proposed by [30].

The optimization algorithms SGD and SGD with Nesterov were used, along with the Cross-Entropy loss function. Table 5 summarizes the parameters used during the training phase.

The objective of the training is that Light CNN-29 learns to extract representative features from facial images and not only to classify them. In this way, Light CNN-29 can be applied to other databases to extract features from facial images to be used as input for similarity functions. Thus, all the following processes make use of the 256-dimensional feature set obtained by Light CNN-29. Table 6 shows the results achieved by the original model and the models trained on the LWIR spectral band, using as similarity function the cosine similarity.

With the results achieved, it is seen that the fine-tuning allowed the network to learn to extract more representative features of facial images of the LWIR spectral band. It is also noticeable that the model that achieved the best results was the SGD without Nesterov, which was chosen for the remaining experiments.

#### 5.5.2. Similarity Functions and Score-Level Fusion

At this stage, we have three Light CNN-29 models, each responsible for extracting features from a specific band. Only the Light CNN-29 responsible for the extraction of features from the LWIR spectral band underwent a fine-tuning. To proceed with classification, it was necessary to find the similarity function that best fits the face recognition task.

Table 7 present the results achieved with the similarity functions cosine similarity and Euclidean distance. The results show that the cosine similarity function is the one that obtains the best score, which is in agreement with [34,35].

It is now possible to use the scores obtained by each spectral band to proceed to the final classification. A fusion of the achieved scores was performed using (1). Two studies were conducted, with different weights of each band (*W_b_* of Equation (1)) as shown in Table 8 and Table 9.

In study 1, the previously obtained test results are not considered; thus, the same weight is used in all spectral bands. The final score is a simple arithmetic mean of the scores of the individual bands, which assumes that all spectral bands have the same classification capacity.

The *W_b_* values in study 2 are derived from the mean of the Rank-1 average precision of each of the spectral bands in the tests performed on the TUFTS-Pose, TUFTS-Exp and CASIA NIR-VIS 2.0 databases (results obtained with the cosine similarity function in Table 7) rounded to tenths. Thus, in study 2, the final score was obtained as weighted arithmetic mean, where each band presents different weights reflecting its classification accuracy.

Table 9, Table 10 and Table 11 show our final face recognition results using both the individual bands and the combination of bands with the two different weight sets (Study 1 and Study 2).

Table 9 presents the results obtained with the TUFTS-Pose database. These results show that study 2 achieved better results than study 1, in the Rank-1 and Rank-3 metrics by 0.1 percentage points, and the TAR@FAR = 0.001 metric by 3 percentage points. The superiority of the results obtained by study 2 compared to study 1 shows that the weight assigned to the LWIR spectral band should be lower than the weight assigned to the others because the characteristics obtained in the LWIR spectral band are the least representative of the identity.

Analyzing the results of the different spectral bands separately, the NIR spectral band achieved the best results due to its robustness towards the variation of illumination present in the TUFTS-Pose database. Despite the promising results of the NIR band when used solo, study 2 obtained superior results in all metrics, with particular emphasis on Rank-1 (from 99.0% to 99.5%) and TAR@FAR = 0.001 (from 93.1% to 93.5%). It is relevant to point out that only the results obtained with score fusion reached the 100% accuracy rate in the assessed Ranks (Rank-4 for study 1 and Rank-3 for study 2).

Table 10 shows the results obtained with the TUFTS-Exp database. An analysis of the results allows us to see that the face recognition results obtained are better with score fusion, where both studies obtained the same result as the VIS spectral band in Rank-1 (99.6%) but managed to achieve a higher result in Rank-2 (100% against 99.6% of the VIS spectral band). However, the best result for TAR@FAR = 0.001 is obtained using only the VIS spectral band, with 99.4%, while the second-best result was obtained in study 2, with 99.3%.

The results achieved using the CASIA NIR-VIS 2.0 database (Table 11) show that study 1 reached a value of 100% in Rank-1. Using the VIS and NIR spectral bands separately, the results were 99.9% and 99.6%, respectively, using the same metric. It should be noted that study 2 was not performed for the CASIA NIR-VIS 2.0 database, as the difference between study 1 and study 2 is the weight assigned to the LWIR spectral band, which it does not have. In the TAR@FAR = 0.001 metric, study 1 matches the result for the VIS spectral band with 100%.

Performing a global analysis of all results, we can observe that the fusion of scores mainly favors cases where the results obtained by the different spectral bands separately were less satisfactory. Looking at the results obtained with the TUFTS-Exp and CASIA-NIR-VIS 2.0 databases (Table 10 and Table 11), it is clear that the VIS spectral band already obtains very high values in all metrics. This fact makes the fusion of scores not so effective. However, despite a decrease of the TAR@FAR = 0.001 in Table 10, the results obtained by the fusion of scores, in general, were higher than those obtained by the spectral bands separately. The results obtained thus demonstrate the benefit of using multi-spectral images in a face recognition system.

## 6. Conclusions

In this paper, a multi-spectral face recognition system in an uncontrolled environment has been proposed, aiming to make a decision with the largest amount of data available, i.e., using the facial images obtained by the different spectral bands. The system is composed of three modules: (i) face detection and alignment, (ii) image synthesis and (iii) face recognition.

The state of the art regarding face recognition systems in an uncontrolled environment has led to the conclusion that image synthesis methods, mainly with GANs, have been used to combat intrapersonal variations, such as the difference in pose and facial expression. On the other hand, in the area of multispectral face recognition, with a plurality of solutions presented by the use of multispectral images, fusion methods are those that make the most use of images captured in different spectral bands in order to make a decision. The main problem encountered is the limited number of images (and people) in multispectral databases in an uncontrolled environment, which makes it challenging to train convolutional neural networks, which are the most used method for feature extraction.

Several techniques were implemented to validate them in different multi-spectral bands, since all of them were trained on visible databases, as well as to analyze the influence of facial image features (pose, illumination and expression). This analysis aimed to select the most appropriate technique for each module of the proposed face recognition system.

For the face detection task, three networks were evaluated qualitatively and quantitatively, which allowed us to conclude that the DSFD network was the most appropriate since it maintained a high accuracy in the different spectral bands. For the landmark detection task, three networks were evaluated qualitatively, and it was concluded that the 2D-FAN network was the best fit due to its ability to correctly identify facial landmarks in different spectral bands with a diversity of facial poses. Such evaluations allowed us to select the methods that are best suited for these tasks with multispectral images in an uncontrolled environment. Thus, this work presents an efficient face detection and face alignment module for a multispectral face recognition system in an uncontrolled environment.

The present work also performed evaluations of different face normalization methods, through image synthesis, to produce face images with a frontal pose. The FFWM and FNM models were analyzed, where the FNM model produced the most realistic facial images for the visible and NIR spectral bands, maintaining the proportions of the face and the most relevant facial features. Further analysis of the FNM model allowed us to conclude that: (i) the greater the pose variation, the greater the advantage in using the FNM model and (ii), the NIR images allow us to obtain a better identification/verification than the visible images because pose variation can entail variations in illumination, to which the NIR band is resistant.

The analysis of the performance of the different models allowed the selection of the most suitable one for a multispectral face recognition system in an uncontrolled environment, as well as the identification of the most advantageous situations for its use.

The extraction of the feature sets of the facial images from the different spectral bands is performed using Light CNN-29 [30], with a fine adjustment to the network weights for the LWIR spectral band since it was trained on the visible spectral band. For the classification phase, identification is performed in the different spectral bands, each producing different scores for each identity. These scores are computed by the similarity between the feature sets of each identity and the feature set of the input facial image. In this work, two different studies were performed for score fusion, which allowed us to conclude that: (i) simply using the different spectral bands to identify is advantageous (study 1) and (ii) a weighted average is beneficial when the different classifiers (of each spectral band) have different levels of reliability (study 2).

On the multi-spectral TUFTS database, with pose variation and expression variation, the results obtained in Rank-1 by the proposed system and with score fusion with a weighted average (study 2) were 99.5% and 99.6%, with the best results obtained using only one spectral band being 99.0% and 99.6%. On the TAR@FAR = 0.001 metric, the results obtained by weighted average are 93.5% and 99.3%, while with only one spectral band 93.1% and 99.4% were obtained. In the CASIA NIR-VIS 2.0 database, score fusion achieved the results of 100.0% in the Rank-1 and TAR@FAR = 0.001 metrics, where without score fusion, 99.9% and 100.0% in Rank-1 and TAR@FAR = 0.001, respectively, are obtained as the best result.

The original contributions of this work include the analysis of several techniques for different tasks, which allowed: (i) the presentation of an efficient face detection and alignment module to be used by any multi-spectral face analysis system, (ii) the identification of the situations in which the FNM model should be used to normalize facial images and (iii) the selection of a similarity function and the weights to be used in the fusion of scores to identify/verify identities. From the experimental results, it is also concluded that the proposed system allows us to obtain high results in multi-spectral face recognition in an uncontrolled environment, where the use of the scores obtained from different spectral bands allows us, in general, to achieve results that are superior to using only the scores obtained by one spectral band.

After performing the work described in this paper, the authors suggest as future work several relevant hypotheses. The first suggestion consists of the creation of a multispectral database to overcome the limitations in the public multispectral databases that currently exist. The second suggestion is to create a prototype and put it to work for access control in high security areas. The third suggestion for future work consists of the adaptation of the image input, to be able to process images obtained by drones with cameras in the spectrum of visible, NIR, SWIR and LWIR, having as an objective the processing of images in real time.

## Figures and Tables

**Figure 1 sensors-22-04219-f001:**
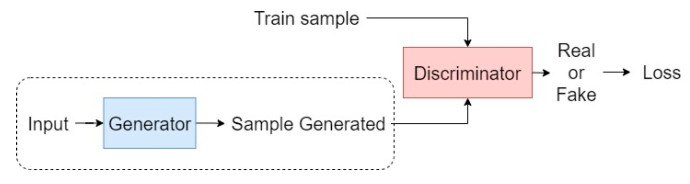
Schematic of the training of a GAN. The dashed line shows the process of sample generation.

**Figure 2 sensors-22-04219-f002:**
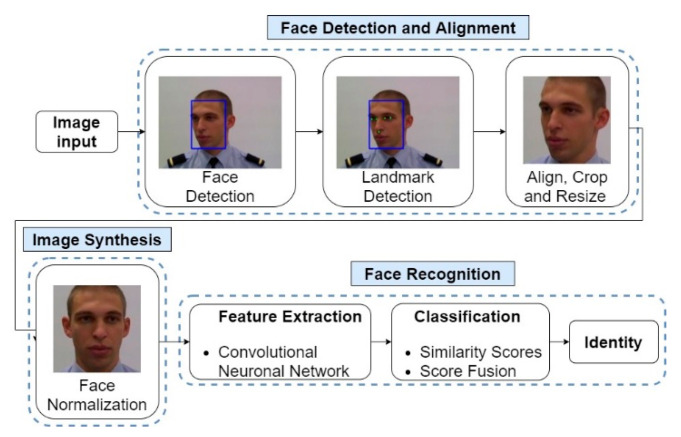
Schematic of the operation of the proposed face recognition system.

**Figure 3 sensors-22-04219-f003:**
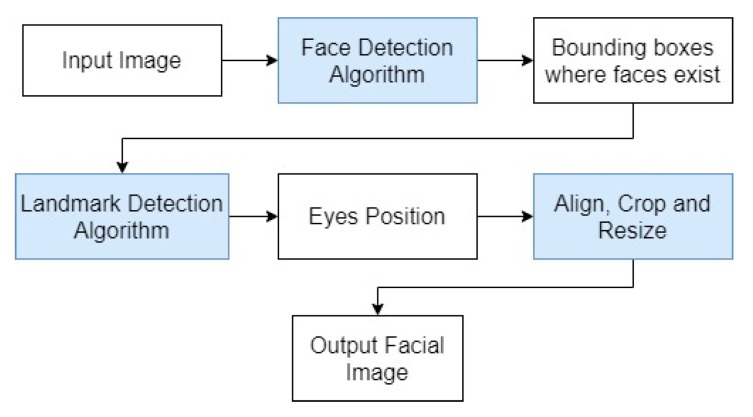
Flowchart of the steps of a facial detection and alignment module.

**Figure 4 sensors-22-04219-f004:**
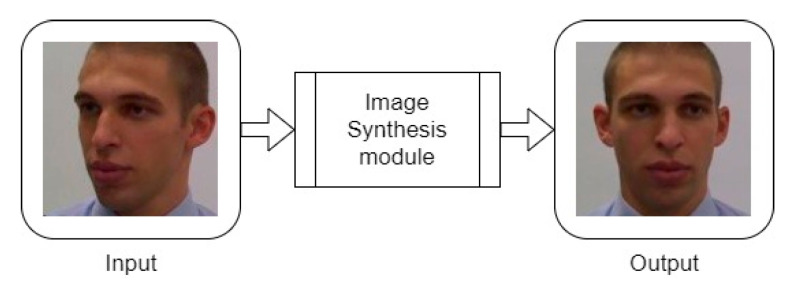
Input and output of the Image Synthesis module (intended function, not the result of a real experiment).

**Figure 5 sensors-22-04219-f005:**
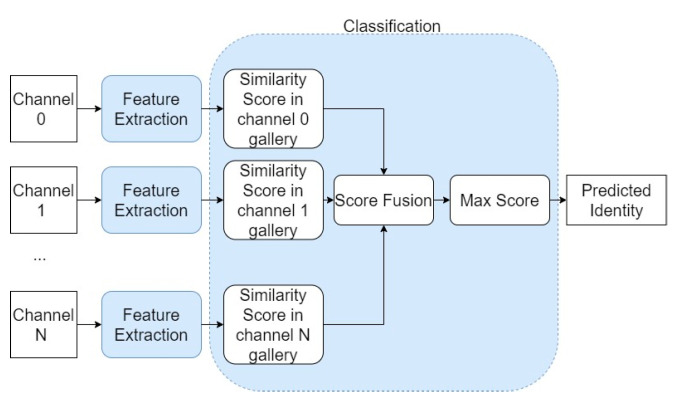
Schematic of the Face Recognition Module.

**Figure 6 sensors-22-04219-f006:**
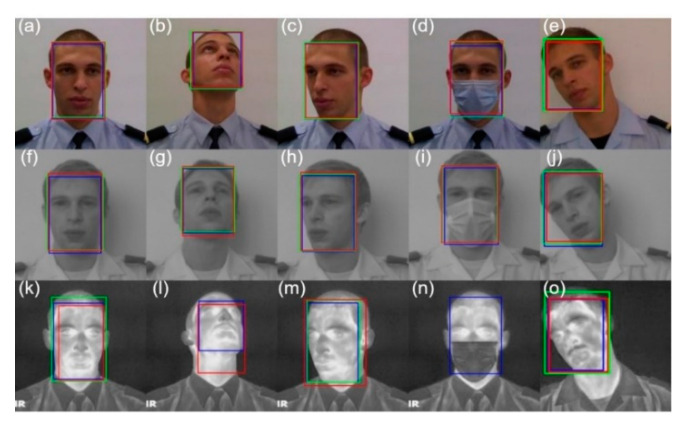
Results obtained by facial detection methods in the spectral bands of VIS (**a**–**e**), NIR (**f**–**j**) and LWIR (**k**–**o**). S3FD—red, DSFD—blue, OpenCV—green.

**Figure 7 sensors-22-04219-f007:**
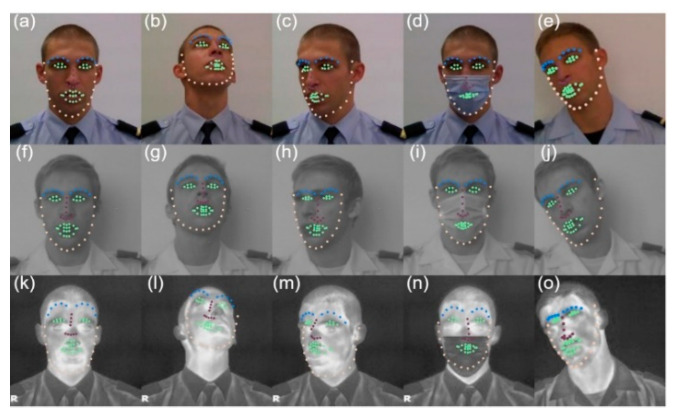
Results achieved by DLIB in the spectral bands of VIS (**a**–**e**), NIR (**f**–**j**) and LWIR (**k**–**o**). Yellow—jawline, green—eyes and mouth, purple—nose, blue—eyebrows.

**Figure 8 sensors-22-04219-f008:**
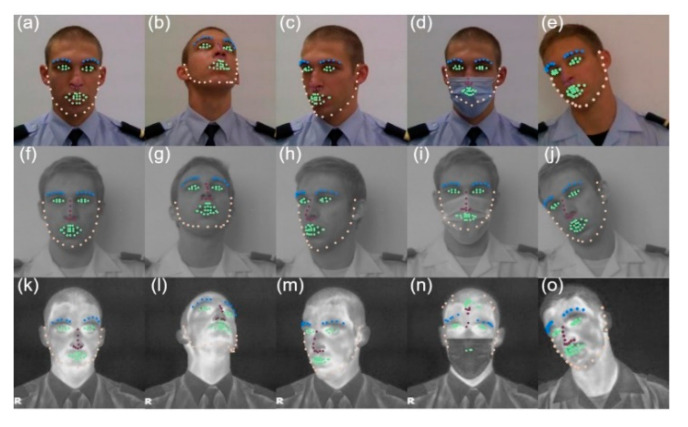
Results achieved by 2D-FAN in the spectral bands of VIS (**a**–**e**), NIR (**f**–**j**) and LWIR (**k**–**o**). Yellow—jawline, green—eyes and mouth, purple—nose, blue—eyebrows.

**Figure 9 sensors-22-04219-f009:**
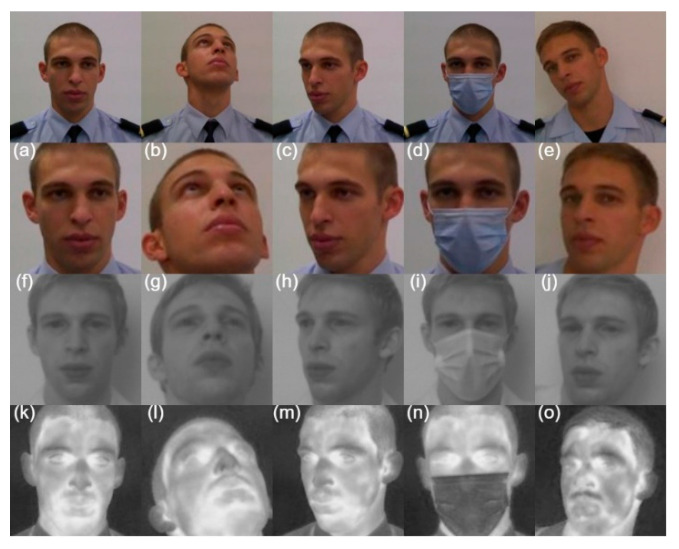
Results achieved by the proposed facial detection and alignment module in the different spectral bands. The images on the top are the originals in the VIS, before processing. Remain images correspond to facial alignment and detection in the spectral bands of VIS (**a**–**e**), NIR (**f**–**j**) and LWIR (**k**–**o**).

**Figure 10 sensors-22-04219-f010:**
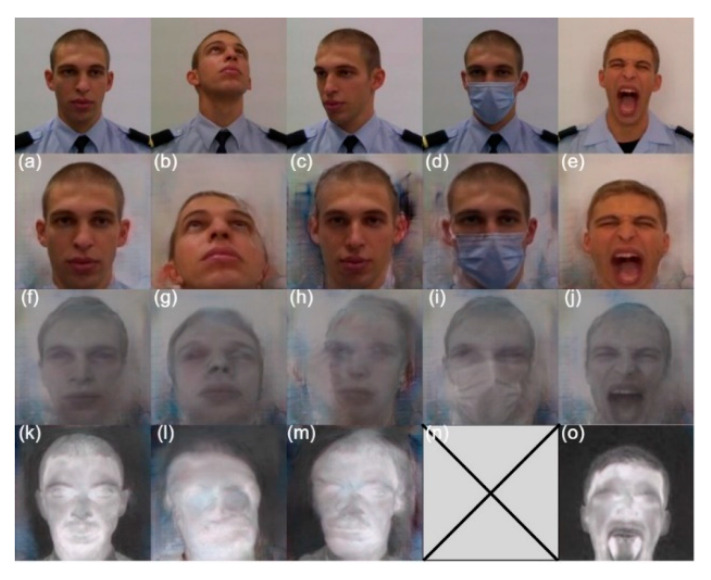
Results achieved by the FFWM in the different spectral bands. The images on the top are the originals in the VIS. The images (**a**–**e**), (**f**–**j**), and (**k**–**o**) were generated by the proposed methodology when it receives as input the images from the VIS, NIR and LWIR bands, respectively.

**Figure 11 sensors-22-04219-f011:**
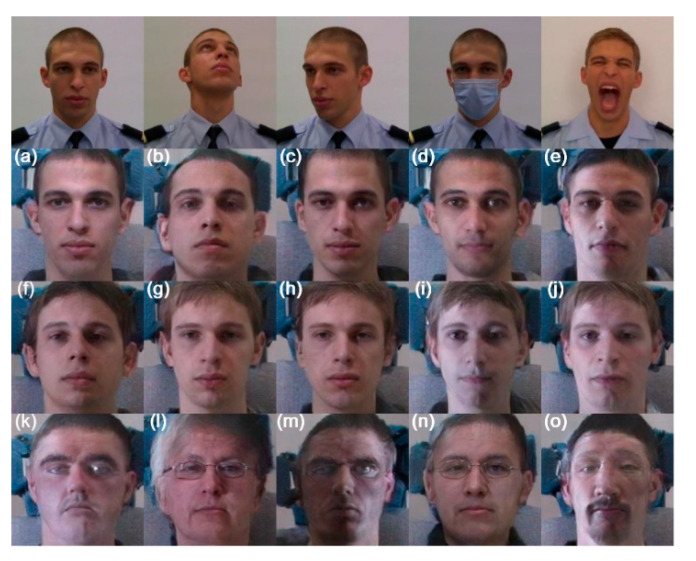
Results achieved by the FNM in the different spectral bands. The images on the top are the originals in the VIS. The images (**a**–**e**), (**f**–**j**), and (**k**–**o**) were generated by the proposed methodology when it receives as input the images from the VIS, NIR and LWIR bands, respectively.

**Table 1 sensors-22-04219-t001:** Accuracy of the different face detection algorithms in the TUFTS database.

Method	Accuracy at Different Spectral Bands (%)		
	VIS	NIR	LWIR
OpenCV	99.2	90.4	77.7
S3FD	99.9	100.0	90.8
DSFD	99.9	100.0	98.8

**Table 2 sensors-22-04219-t002:** Results (in %) with and without FNM on the TUFTS-Pose database, using the Light CNN-29 and cosine similarity score.

	Rank-1		Rank-5		TAR @FAR = 0.001	
	w/o	w/	w/o	w/	w/o	w/
VIS	80.3	96.2	91.0	99.5	60.8	87.2
NIR	98.3	99.0	99.5	99.8	90.4	91.9
LWIR	41.8	34.9	58.2	57.8	28.7	14.0

**Table 3 sensors-22-04219-t003:** Results (in %) with and without FNM on the TUFTS-Exp database, using the Light CNN-29 and cosine similarity score.

	Rank-1		Rank-5		TAR @FAR = 0.001	
	w/o	w/	w/o	w/	w/o	w/
VIS	99.6	93.3	100.0	98.5	99.4	82.9
LWIR	67.5	42.7	83.3	48.2	57.0	23.9

**Table 4 sensors-22-04219-t004:** Results (in %) of rank-1 with and without FNM on TUFTS-Pose database with quantification of pose variation, using the Light CNN-29 and cosine similarity score.

		Pose Variation			
		±60°	±45°	±30°	±15°
VIS	w/o	43.3	77.5	100.0	100.0
	w/	87.4	97.7	99.5	100.0
NIR	w/o	93.4	99.7	100.0	100.0
	w/	96.5	99.4	100.0	100.0

**Table 5 sensors-22-04219-t005:** Parameters used in the training procedure.

Parameter	Value
Batch Size	16
Learning Rate	10^−4^
Momentum	0.9
Epoch Number	10

**Table 6 sensors-22-04219-t006:** Rank-1 results (in %) achieved by different models for extraction of LWIR band features.

	Original	SGD	SGD Nesterov
TUFTS-Pose	41.8	55.5	54.3
TUFTS-Exp	67.5	79.6	75.9

**Table 7 sensors-22-04219-t007:** Rank-1 results (in %) achieved in the face recognition task with the cosine similarity (CSim) and Euclidean Distance (EDis).

	TUFTS-Pose		TUFTS-Exp		CASIA NIR-VIS 2.0	
	CSim	EDis	CSim	EDis	CSim	EDis
VIS	96.2	95.3	99.6	99.4	99.9	99.8
NIR	99.0	96.6	-	-	99.3	99.1
LWIR	55.5	42.0	79.6	69.6	-	-

**Table 8 sensors-22-04219-t008:** *W_b_* values to be used for each spectral band in the different studies.

	Study 1	Study 2
VIS	1.0	1.0
NIR	1.0	1.0
LWIR	1.0	0.7

**Table 9 sensors-22-04219-t009:** Results (in %) obtained in the face recognition task, in the TUFTS-Pose database.

	Rank					TAR @FAR = 0.001
	1	2	3	4	5	
Study1	99.4	99.8	99.9	100.0	100.0	90.5
Study2	99.5	99.8	100.0	100.0	100.0	93.5
VIS	96.2	98.7	99.1	99.4	99.5	87.4
NIR	99.0	99.7	99.7	99.8	99.8	93.1
LWIR	55.6	62.2	66.7	69.9	72.6	30.5

**Table 10 sensors-22-04219-t010:** Results (in %) achieved in the face recognition task, using the TUFTS-Exp database.

	Rank					TAR @FAR = 0.001
	1	2	3	4	5	
Study1	99.6	100.0	100.0	100.0	100.0	98.7
Study2	99.6	100.0	100.0	100.0	100.0	99.3
VIS	99.6	99.6	99.8	100.0	100.0	99.4
LWIR	79.6	86.3	88.5	90.4	91.6	54.9

**Table 11 sensors-22-04219-t011:** Results (in %) achieved in the face recognition task, using the CASIA NIR-VIS 2.0 database.

	Rank					TAR @FAR = 0.001
	1	2	3	4	5	
Study1	100.0	100.0	100.0	100.0	100.0	100.0
VIS	99.9	100.0	100.0	100.0	100.0	100.0
NIR	99.6	99.7	99.9	99.9	99.9	99.1

## Data Availability

Not applicable.

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
