# Peer review of "Multispectral Facial Recognition in the Wild"

_sensors, 2022, doi:10.3390/s22114219_

Round 1
Reviewer 1 Report
This is a novel and interesting study, with promising results.
The authors proposed a multi-spectral face recognition system in an uncontrolled environment for the identification/authentication of people through their facial images.
The paper is well-written, all explanations and conclusions are clear and supported by computational data. The English language and style are fine. Therefore, I recommend accepting this paper in a present form.
Author Response
"This is a novel and interesting study, with promising results.
The authors proposed a multi-spectral face recognition system in an uncontrolled environment for the identification/authentication of people through their facial images.
The paper is well-written, all explanations and conclusions are clear and supported by computational data. The English language and style are fine. Therefore, I recommend accepting this paper in a present form."
Reply to reviewer comments:
The authors are grateful for the words of encouragement written by the reviewer.
We took this opportunity to read the paper and correct some typos that we detected after the paper was submitted.
Reviewer 2 Report
The paper is difficult read. It provides an overview of multi-spectral facial recognition.
- It is not clear if the authors are inspired by the work of others or apply it in new environment.
- To give an example. It is not clear how successful is the localisation of faces. In my smart phone the system localizes faces by putting green boxes around them.
- In the image synthesis the authors rotate side view of a face to a frontal face. How successful is this algorithm
- The topic facial expressions are hardly discussed, but reduced to neutral faces.
- We prefer to see test results from specified multi-spectral methods in specified situations.
- In 5.5.1 the authors decided to make a fine adjustment, why?
- To summarize the line of the paper is not clear and the results and context are not specified.
Author Response
Comment 1:
"The paper is difficult read. It provides an overview of multi-spectral facial recognition. "
"1. It is not clear if the authors are inspired by the work of others or apply it in new environment."
Reply to reviewer comment 1:
The authors were inspired by the works of others, and modified those works for this new environment.
Comment 2:
"2. To give an example. It is not clear how successful is the localisation of faces. In my smart phone the system localizes faces by putting green boxes around them.”
Reply to reviewer comment 2:
The face localization algorithm provides an area where it believes the face is in the image. It is considered a correct location when the algorithm returns this area correctly. For testing purposes, the algorithm places the perimeter of the area with a color, as seen in figure 6, to facilitate visual inspection of the results obtained by it. Once this phase is finished, the landmark’s location, image synthesis, and finally face recognition follow.
Comment 3:
"3. In the image synthesis the authors rotate side view of a face to a frontal face. How successful is this algorithm"
Reply to reviewer comment 3:
A typo was detected in Table 2, with switched labels. This error has been corrected. It can now be observed that the use of this algorithm increases the face recognition accuracy in the different metrics for the visible and Near Infrared bands. This leads to the conclusion that the use of this algorithm has been beneficial for these spectral bands.
Comment 4:
"4. The topic facial expressions are hardly discussed, but reduced to neutral faces. "
Reply to reviewer comment 4:
The proposed methodology was tested on a database with different facial expressions (TUFTS-Exp), where we obtained the results presented in different tables, as in tables 3 and 10.
The work is not aimed at recognizing facial expressions per se, but rather facial recognition in scenarios where the person does not cooperate in acquiring the image.
Comment 5:
"5. We prefer to see test results from specified multi-spectral methods in specified situations.”
Reply to reviewer comment 5:
If cameras are available in different spectral bands, facial recognition will be more robust, because each spectral band has its own characteristics. An example of this is the ability to overcome the limitations caused by fog in Near Infrared and plastic materials in Long-wavelength infrared.
Comment 6:
"6. In 5.5.1 the authors decided to make a fine adjustment, why?”
Reply to reviewer comment 6:
It was decided to perform a fine-tuning for the LWIR spectral band because the Light CNN-29 network produced considerably lower results in this band, as the network was trained in the visible spectrum (Table 2 and Table 3). Thus, the fine-tuning allows the network to learn to extract more representative features from facial images in the LWIR spectral band, which led to the results in Table 6.
Comment 7:
"7. To summarize the line of the paper is not clear and the results and context are not specified."
Reply to reviewer comment 7:
With the previous answers and the changes made to the paper, the authors hope to have made the results clearer and easier to understand.
Reviewer 3 Report
|
Line |
Comment/Suggested edit |
|
Title |
“on” the wild --> in? |
|
58-60 |
Revise the end of the sentence “and public multispectral databases is performed.” |
|
118 and 119 |
“consists in …” --> consists of |
|
268 |
Split the sentences. “its identity the similarity…” --> “its identity. The similarity…” |
|
358 |
Figure 10 is separating the sub-heading from the text. |
|
450 |
“classification was” --> classification, it was |
|
550, 553, 554 |
“TAV@TAF=0.001” |
The manuscript clearly highlights its contributions and validates the proposed multispectral face recognition approach via quantitative analysis and comparisons. The manuscript is well structured, coherent, and builds on the state-of-the-art to address an identified gap in face recognition in the wild, mainly illumination variation. However, there are some points to consider:
It would be useful if the introduction/background section included samples of images in the different spectral bands. Also samples from TUFTS and CASIA NIR VIS 2.0 to showcase how the images resemble the uncontrolled environments characteristics mentioned in the manuscript.
It is unclear what “better results” mean in assessing the facial landmarks detectors and whether the choice of 2D-FAN was based solely on visual inspection of the points, or via a similarity measure with respect to the VIS results. Similarly with the decision between the synthesisers. It appears the normalisation models’ results are greatly influenced by their training data, which is understandable, and is evident by noticeable changes in the normalised face from the original and the presence of false artefacts like glasses and facial hair.
The manuscript does indicate the use of qualitative data to assist models’ choice, but it appears this is limited to the presented cases? A more solid quantitative measure is required here to support these claims and generalise the conclusion by drawing some metrics (perhaps thresholded distance from VIS results) over a large population of images.
Tables 2-4. describes “results” while it would be useful to clearly state it is the Cosine similarity score.
Analysis of Table 2 should be revised. Lines 379-378 state that Rank-1 scores improve with the use of FNM for both VIS and NIR, but according to the table, this is only true for NIR. The VIS score for Rank-1 dropped by %15.9 with FNM and dropped by %8.5 for Rank-5 with FNM.
“In the remaining metrics, it is also observed better values with the use of the normalization model.” Line 399. What kind of metrics is meant here? If this is referring to the Rank-5 and FAR, then again, this conclusion is inconsistent with the data presented in the table.
The same remark applies to the analysis of Table 3 in lines 408-409. Perhaps the columns (w/ and w/o) are incorrectly labelled because it shows that with (w/) FNM, the results are better for LWIR.
The explanation given for the additional layer in the LWIR band network with respect to training images being different from testing images is not clear.
Limitations and future work section should conclude.
Understandably data availability is an issue as highlighted, but one may argue the terms “in the wild” and “uncontrolled environment” are used very loosely here since the used image sets are still controlled to a large extent and may not resemble in-the-wild examples from other computer vision problems.
Author Response
Comment 1:
|
Line |
Comment/Suggested edit |
|
Title |
“on” the wild --> in? |
|
58-60 |
Revise the end of the sentence “and public multispectral databases is performed.” |
|
118 and 119 |
“consists in …” --> consists of |
|
268 |
Split the sentences. “its identity the similarity…” --> “its identity. The similarity…” |
|
358 |
Figure 10 is separating the sub-heading from the text. |
|
450 |
“classification was” --> classification, it was |
|
550, 553, 554 |
“TAV@TAF=0.001” |
Reply to reviewer comment 1:
The authors are grateful for the typo corrections written by the reviewer. These corrections, as well as others detected by the authors were made in the paper.
Comment 2:
"It would be useful if the introduction/background section included samples of images in the different spectral bands. Also samples from TUFTS and CASIA NIR VIS 2.0 to showcase how the images resemble the uncontrolled environments characteristics mentioned in the manuscript."
Reply to reviewer comment 2:
It was our concern to include only images produced by us or acquired by our cameras, and not to put images obtained by third parties to avoid publishing images that may have copyrights or may need express authorization from the author of the image, to be published.
Comment 3:
"It is unclear what “better results” mean in assessing the facial landmarks detectors and whether the choice of 2D-FAN was based solely on visual inspection of the points, or via a similarity measure with respect to the VIS results. Similarly with the decision between the synthesisers. It appears the normalisation models’ results are greatly influenced by their training data, which is understandable, and is evident by noticeable changes in the normalised face from the original and the presence of false artefacts like glasses and facial hair. "
Reply to reviewer comment 3:
Regarding the first question, the choice of 2D-FAN was based on visual inspection of the points. Also, in the state of the art it was possible to verify that this network was more often used to detect landmarks in an uncontrolled environment.
Regarding the second question, the training data has a lot of influence on this type of networks, thus, if the input image is in the LWIR band, (quite different from the training data), the network synthesizes an image with false artifacts.
Comment 4:
" The manuscript does indicate the use of qualitative data to assist models’ choice, but it appears this is limited to the presented cases? A more solid quantitative measure is required here to support these claims and generalise the conclusion by drawing some metrics (perhaps thresholded distance from VIS results) over a large population of images."
Reply to reviewer comment 4:
For the choice of landmark detection algorithm, 2D-FAN was chosen based on the state-of-the-art analysis, as well as through qualitative results obtained by us. The results presented here are a few examples chosen by the authors from a larger universe.
The same applies to the choice of image synthesizer. The choices were made considering the results presented in the papers of each work, the way they operated, the training data, and finally, a qualitative evaluation made by us, with more images than the ones presented here.
Comment 5:
"Tables 2-4. describes “results” while it would be useful to clearly state it is the Cosine similarity score. "
Reply to reviewer comment 5:
We have made that change, as well as adding the fact that it is done with the Light CNN-29 network.
Comment 6:
"Analysis of Table 2 should be revised. Lines 379-378 state that Rank-1 scores improve with the use of FNM for both VIS and NIR, but according to the table, this is only true for NIR. The VIS score for Rank-1 dropped by %15.9 with FNM and dropped by %8.5 for Rank-5 with FNM. "
Comment 7:
"In the remaining metrics, it is also observed better values with the use of the normalization model.” Line 399. What kind of metrics is meant here? If this is referring to the Rank-5 and FAR, then again, this conclusion is inconsistent with the data presented in the table.
The same remark applies to the analysis of Table 3 in lines 408-409. Perhaps the columns (w/ and w/o) are incorrectly labelled because it shows that with (w/) FNM, the results are better for LWIR. "
Reply to reviewer comment 6 and 7:
Thank you for the repair. There is indeed the error pointed out. Tables 2 and 3 have been corrected.
Comment 8:
"The explanation given for the additional layer in the LWIR band network with respect to training images being different from testing images is not clear. "
Reply to reviewer comment 8:
This last layer is used as the input of the softmax cost function and is simply set to the number of training set identities, as proposed by the authors of the Light CNN-29 (the network that we are adding the last layer). We updated the paper with this information.
Comment 9:
"Limitations and future work section should conclude."
Reply to reviewer comment 9:
We appreciate the suggestion indicated by the reviewer to suggest future work that goes beyond the limitations that the authors encountered in carrying out the present work. We have inserted the following text in the paper:
"After performing the work described in this paper, the authors suggest several pertinent directions for future work. First, the creation of a multispectral database would help overcome the gaps in the currently existing public multispectral databases. Second, creating a prototype to test the proposed methods for access control in high security areas would allow us to better assess the practical applicability of the method. Finally, a conclusive validation of facial recognition in the wild would be possible by deploying the system in drones with cameras in the visible, NIR, SWIR and LWIR spectrum, with the goal of processing images in real time."
Comment 10:
"Understandably data availability is an issue as highlighted, but one may argue the terms “in the wild” and “uncontrolled environment” are used very loosely here since the used image sets are still controlled to a large extent and may not resemble in-the-wild examples from other computer vision problems."
Reply to reviewer comment 10:
The public datasets used are available on the internet, so our limitation is due to the limited variety of images. We tried to minimize this obstacle by acquiring images at our institution in various poses.
Reviewer 4 Report
A multi-spectral face recognition system is a solution that promises to achieve better results in facial recognition, the subject adressed in this paper being interesting and relevant in this field.
At a more detailed analysis we identified the following minor problems:
- The values shown in Tables 2 and 3 should be checked, as the values in the columns with (w /) and without (w / o) the use of FNM seem to be interchanged.
- Original contributions need to be more clearly emphasized.
In conclusion, I recommend accepting the article after minor revisions, in order to address the above observations.
Author Response
Comment 1:
"- The values shown in Tables 2 and 3 should be checked, as the values in the columns with (w /) and without (w / o) the use of FNM seem to be interchanged. "
Reply to reviewer comment 1:
Thank you for the repair. There is indeed the error pointed out. Tables 2 and 3 have been corrected.
Comment 2:
"- Original contributions need to be more clearly emphasized. "
Reply to reviewer comment 2:
We thank the reviewer for his comment and have revised the paragraph where we present the original contributions. In this way, the contributions of this paper have become more explicit and objective.
Comment 3:
"In conclusion, I recommend accepting the article after minor revisions, in order to address the above observations."
Reply to reviewer comment 3:
The authors are grateful for the words of encouragement written by the reviewer.
We took this opportunity to read the paper and correct some typos that we detected after the paper was submitted.
Round 2
Reviewer 2 Report
The paper has only a few innovative aspects and adapts the work of others.
The authors discussed our comment marginally, so we are not satisfied
Author Response
Concern 1:
- It is not clear if the authors are inspired by the work of others or apply it in new environment.
Authors response:
The proposed system combines, in a novel way, and applies, to a novel problem, a set of algorithms for facial analysis and machine learning from others. Three face detection algorithms (S3FD, OpenCV NN and DSFD - manuscript v3 lines 209 to 226) are explored and evaluated in their ability to detect faces in multispectral images. Our study concluded that DSFD maintains a very high accuracy for the different spectral bands, even in the LWIR spectral band, being the best one for face detection in a multispectral facial analysis system.
Two image synthesis methods (FNM and FFWM - manuscript v3 lines 229 to 251) were evaluated in the image normalization process (generate frontal views and neutral expressions) on different multispectral bands. Our study concluded that, in spite of both networks were unable to provide good results in LWIR band, (i) the FNM model produced the most realistic facial images for the Visible and NIR spectral bands regarding the generation of frontal views and (ii) the greater the pose variation, the greater the advantage in using the FNM model. This study showed as well that NIR images allow obtaining a better identification/verification than the Visible images with pose variation (1st column of Table 2), because pose variation can entail variations in illumination, to which the NIR band is resistant.
Finally, in the Face Recognition Task, we explored (manuscript section 5.5.2) two similarity functions to obtain similarity score of the features extracted (Table 7), as well the results obtained by each spectral band and combined with score fusion (Table 9, 10 and 11). Our study concluded that the best similarity function to the face recognition is the cosine similarity and, regarding the score fusion, that: (i) the use of the different spectral bands combined strengthens the face recognition, allowing the classifier to use more data available, with each spectral band providing different features of the input facial image and (ii) a weighted average in the score fusion is beneficial when the different classifiers (of each spectral band at solo) have different levels of reliability.
Thus, despite the individual technologies are from others, their combination and evaluation in a new problem is novel. We believe the conclusions presented in the paper are very valuable to the community as they indicate which models and combinations work best in the different datasets and spectral bands both in the individual processing steps, as well as in the final task of face recognition.
Authors action:
The conclusions were reviewed and updated, now have the following text:
“For the face detection task, three networks were evaluated qualitatively and quantitatively, which allowed concluding that the DSFD network was the most appropriate since it maintained a high accuracy in the different spectral bands. For the landmark detection task, three networks were evaluated qualitatively and was concluded that the 2D-FAN network was the best fit due to its ability to correctly identify facial landmarks in different spectral bands with a diversity of facial poses. Such evaluations allowed selecting the methods that are best suited for these tasks with multispectral images in an uncontrolled environment. Thus, this work presents an efficient face detection and face alignment module for a multispectral face recognition system in an uncontrolled environment.
The present work also performed evaluations of different face normalization methods, through image synthesis, to produce face images with a frontal pose. The FFWM and FNM models were analyzed, where the FNM model produced the most realistic facial images for the Visible and NIR spectral bands, maintaining the proportions of the face and the most relevant facial features. Further analysis of the FNM model allowed us to conclude that: (i) the greater the pose variation, the greater the advantage in using the FNM model and (ii), the NIR images allow obtaining a better identification/verification than the Visible images because pose variation can entail variations in illumination, to which the NIR band is resistant.
The analysis of the performance of the different models allowed the selection of the most suitable one for a multispectral face recognition system in an uncontrolled environment, as well as the identification of the most advantageous situations for its use.”
“The original contributions from this work includes the analysis of several techniques for different tasks , which allowed (…)”
Comment 2:
- To give an example. It is not clear how successful is the localisation of faces. In my smart phone the system localizes faces by putting green boxes around them.
Authors response:
The face localization algorithm provides an area where it believes the face is in the image. It is considered a correct location when the algorithm returns this area correctly. For testing purposes, the algorithm places the perimeter of the area with a color, as seen in figure 6, to facilitate visual inspection of the results. Regarding quantitative results, presented in Table 1, it can be observed that the DSFD method (the one selected) achieves a precision of 99.9% in VIS, 100.0% in NIR and 98.8% in LWIR, values which are considered high in the task of localization of faces.
Authors action:
Conclusions were updated (manuscript v3 lines 551 to 559):
“For the face detection task, three networks were evaluated qualitatively and quantitatively, which allowed concluding that the DSFD network was the most appropriate since it maintained a high accuracy in the different spectral bands. For the landmark detection task, three networks were evaluated qualitatively and was concluded that the 2D-FAN network was the best fit due to its ability to correctly identify facial landmarks in different spectral bands with a diversity of facial poses. Such evaluations allowed selecting the methods that are best suited for these tasks with multispectral images in an uncontrolled environment. Thus, this work presents an efficient face detection and face alignment module for a multispectral face recognition system in an uncontrolled environment.”
Comment 3:
- In the image synthesis the authors rotate side view of a face to a frontal face. How successful is this algorithm.
Authors response:
The evaluation is done qualitatively (Figures 10 and 11), and quantitatively (Tables 2 and 4). The qualitative analysis allows us to see that the majority of the time the face is correctly transformed to the frontal perspective, but that the identity may not be preserved. To this end, the quantitative analysis evaluates the quality of the person's identity identification before (w/o) and after the synthesis (w/), always verifying improvements except in the LWIR. For reference, it is seen in Table 2 that the use of image synthesis improves the rank-1 results in the visible band from 80.3% to 96.2% and in the NIR band from 98.3% to 99.0%.
Authors action:
Legends of tables 2, 3 and 4 were updated. Column titles of tables 2 and 3 were corrected.
Comment 4:
- The topic facial expressions are hardly discussed, but reduced to neutral faces.
Authors response:
The work is not aimed at recognizing facial expressions per se, but rather facial recognition in scenarios where the person does not cooperate in acquiring the image. Because of this, facial expressions were not explicitly analyzed as they were not an objective of the paper. However, the proposed methodology was tested on a database with different facial expressions (TUFTS-Exp), where we obtained the results presented in different tables, as in tables 3 and 10.
As we can see from table 10, the results obtained by the spectral bands solo (not used in fusion) are already quite high (99.6% in the case of VIS), as current face recognition systems can already overcome with some ease this characteristic of the uncontrolled environment, especially in VIS, when compared to the variation of facial pose. However, it can be observed that the multispectral recognition (study 1 and 2), the results obtained in rank-2 are 100.0%, while the best spectral band used in solo (VIS) can only obtain this result in rank-4, thus demonstrating the benefit of the proposed methodology by using different spectral bands with score fusion.
Authors action:
Sentence updated (manuscript v3 line 496)
Comment 5:
- We prefer to see test results from specified multi-spectral methods in specified situations.
Authors response:
The overall goal is to perform facial recognition of a person in a non-collaborative (in the wild) scenario. The specific situation most studied in this work is a photograph of a person when they are not facing the camera, because it is the extreme situation for facial recognition, since it does not allow the visualization of the face completely. The proposed methodology starts by performing face detection, followed by rotating the face to face the camera, followed by face recognition. To go further than most other works, we used multispectral cameras (Visible, NIR, LWIR) with the advantage of allowing face recognition even when in one of the multispectral bands the face image is hidden. We present the results obtained from 3 databases, with different scenarios.
TUFTS-Pose database presents 9 facial images per person with variations in pose in the horizontal plane between -60° and +60° (our main focus) and different brightness levels (in some cases, the face is in the dark, unable to be seen in the VIS spectral band). In this database, the use of a score fusion with the three available spectral bands (VIS, NIR and LWIR) achieves the best result (Table 9), with 99.5% in the study 2, compared to the 99.0% obtained by NIR (best spectral band when used at solo).
TUFTS-Exp database presents five facial images in a frontal pose per person with different facial expressions, obtained in the Visible and LWIR bands. The results (see table 10) of multispectral recognition (study 1 and study 2) were 100.0% and 100% in rank-2, while the best spectral band used at solo (VIS) can only obtain this result in rank-4.
CASIA NIR-VIS 2.0 database presents images with slight variations in pose, as well as some variations in illumination, where NIR images are captured in the dark. As almost all images are taken with frontal pose, with few features of the uncontrolled environment, the results obtained (Table 11) by the spectral bands in solo are already quite high (99.9%). Still, with score fusion (using the visible and NIR spectral bands together), it allowed us to obtain a result of 100.0%, demonstrating that the use of score fusion is also beneficial in more controlled scenarios.
Authors action:
Updated the conclusions (manuscript v3 lines 577, 578 and also 581)
Comment 6:
- In 5.5.1 the authors decided to make a fine adjustment, why?
Authors response:
Light-CNN 29 is a network that extracts features from facial images, features which are used to determine the similarity score between two different facial images. This feature extraction network was trained on the visible spectral band. Due to the difference between the LWIR and VIS spectral bands (see Figure 6), the features that were extracted in the LWIR spectral band had a low level of representativeness, that is, the features extracted from two facial images of the same identity in the LWIR band did not have as much similarity as in the VIS band (Table 2 and 3). As such, it was decided to perform a fine adjustment, so that the network would learn to extract more representative features. This fine adjustment improved the way facial features were extracted in the LWIR spectral band, improving the results of rank-1 from 41.8% to 54.3% in TUFTS-Pose and from 67.5% to 75.9% in TUFTS-Exp. It should be noted that these results are still well below those obtained in the VIS or NIR spectral bands, where the images obtained are very similar to those obtained in VIS (see Figure 6), because the images in LWIR are also sensitive to the environment and surroundings, as well the emotional, physical and health conditions of the individual. This leads to, in the same conditions of light and facial pose, the features obtained in two different sessions may be different.
Authors action:
To address the question, that may arise to future readers, we added the follow (manuscript v3 lines 431-434)
“(..) it was decided to make only one fine adjustment to the LWIR band feature extraction network, because the results obtained in this band are considerably lower, due to the network having been trained in the visible. Thus, the fine-tuning aims for the network to learn to extract more representative features from facial images in the LWIR spectral band.”
Comment 7:
- To summarize the line of the paper is not clear and the results and context are not specified.
Authors response:
We hope that the answer given in this second round of review have made the results clearer and easier to understand.
Considering that the comment also refers to the lack of objectivity in the context and in the results presented in the conclusions, we also made some changes to this section.
Authors action: We added a paragraph (manuscript v3 lines 537-545) regarding the state of the art, in order to give a context of the problem to be solved, as well as what has been done in the area:
“State of the art regarding face recognition systems in an uncontrolled environment has led to the conclusion that image synthesis methods, mainly with GANs, have been used to combat intrapersonal variations, such as the difference in pose and facial expression. On the other hand, in the area of multispectral face recognition, with a plurality of solutions presented by the use of multispectral images, fusion methods are those that most make use of images captured in different spectral bands in order to make a decision. The main problem encountered is the limited number of images (and people) in multispectral databases in an uncontrolled environment, which makes it challenging to train convolutional neural networks, which are the most used method for feature extraction.”
We also made several changes to better articulate the sequence of the work developed, the intent of each of the steps, and to emphasize the contributions of the work. (lines 556-562, 570-572 and 577-578).
At last, we added a paragraph about future work: (lines 602-609).
“After performing the work described in this paper, the authors suggest as future work several relevant hypotheses. The first suggestion consists in the creation of a multispectral database to overcome the limitations in the public multispectral databases that currently exist. The second suggestion is to create a prototype and put it to work for access control in high security areas. The third suggestion for future work consists in the adaptation of the image input, to be able to process images obtained by drones with cameras in the spectrum of visible, NIR, SWIR and LWIR, having as objective the processing of images in real time.”
Comment 8:
The paper has only a few innovative aspects and adapts the work of others. The authors discussed our comment marginally, so we are not satisfied.
Authors response:
We are grateful for the reviewer's words for agreeing that this work has innovative aspects, which is why we submitted it to Journal Sensors. We emphasize that this work is not a replication of work done by others, but rather looking for a solution to a problem. The main objective of this work is to perform facial recognition combining two approaches: to be performed in non-collaborative images (in which the person is not facing to the camera), and to use multi-spectral images of thee spectral bands (visible, NIR, LWIR). As we could not find a solution available on the internet, we proceeded to develop a solution. The overall problem was divided into small problems and a solution was searched for each of the problems. The biggest challenge was establishing a functional link between the various steps to ensure that the overall methodology works. It was necessary to understand each of the steps, when necessary to make its modification and finally to have the methodology working. To ensure that the methodology has a high performance, several tests were carried out, whether to fine-tune input variables for some of the tasks, or to train and test the methodology as a whole.
Authors action:
As the reviewer was not satisfied with the answers that the authors initially gave, the authors performed an analysis of each reviewer's question and prepared a more complete answer, written in responses #1 to #7 to comments #1 to #7 of the reviewer. We also took the opportunity to review some paragraphs of the article, making it clear and objective, contextualizing the results, making the article easier to read and understand, for future readers.